# Neurodevelopmental Disorders: Past, Present, and Future

**DOI:** 10.3390/children10010031

**Published:** 2022-12-24

**Authors:** Elisa Cainelli, Patrizia Bisiacchi

**Affiliations:** 1Department of General Psychology, University of Padova, 35131 Padova, Italy; 2Padova Neuroscience Center, PNC, 35131 Padova, Italy

**Keywords:** ADHD, specific learning disabilities, autism, cognition, neuropsychology

## Abstract

Recent decades have seen a dramatic increase in neurodevelopmental disorders and the attention paid to them. Since their emergence in the not-so-distant past, some neurodevelopmental disorders have undergone considerable redefinition and, beginning in the 21st century, there has been a massive increase in research. In this paper, we briefly review the history of some of them, address some of the issues that characterize their current management and relationship with neurological pathologies, and share some insights for the future.

## 1. Introduction: The Shift in Perspective

Virtually everyone, even people not involved in mental health, has heard about disorders such as dyslexia, autism, or attention-deficit/hyperactivity disorder (ADHD), thanks to the considerable media coverage they have gained in recent years. This phenomenon reflects the great attention of the scientific community and the public to issues that were hardly considered until about fifty years ago [1]. This change in perspective is probably related to a more global process involving all areas of medicine where, thanks to improvements in biomedical techniques and the reduction in mortality rates of many diseases, the focus has shifted from the survival of the individual to their well-being and quality of life [2,3]. Personal, social, educational, or occupational functioning has thus become a crucial aspect [4], and many practitioners have begun to recognize that alterations have multifactorial causes that often originate in the early years of an individual’s life [5,6,7].

In line with these changes, a new diagnostic category was created in the latest version of the Diagnostic and Statistical Manual of Mental Disorders (DSM-5) that fits perfectly with the new edition’s general approach of representing mental disorders throughout the lifespan [8]. This new category, “Neurodevelopmental disorders,” has replaced the category of “Disorders usually first diagnosed in childhood or adolescence,” where “neurodevelopmental” implies an early origin and a neurobiological correlate, in the establishment of which aversive events during gestation or at birth often play a role [9].

Within this category are disorders that can range from very specific learning limitations to global impairment of social skills and intelligence. Some changes are substantial, such as the inclusion of ADHD in this new category, currently the most common among neurodevelopmental disorders, with an estimated worldwide prevalence of 5% [10]. Major changes have also been made in the criteria for the diagnosis of autism [9].

Although the change in perspective embraces a lifespan point of view, in practice, people with neurodevelopmental disorders must come to terms with the discontinuity between childhood and adulthood. In fact, two orders of problems characterize—at last some—neurodevelopmental disorders: “Where do they come from?” and “Where will they go once they leave childhood?” Indeed, while some disorders, such as intellectual disabilities, have a great historical background, and the transition into adulthood is regulated, other disorders are typical of the modern era and limited to childhood. For example, an ADHD diagnosis in childhood is very frequent, but, as shown by the Global Burden of Disability [11], its incidence reduces drastically in adulthood. Is this a true reduction in the incidence, or does the diagnosis change in another? How many psychiatrists in front of impulsivity, inattention, and hyperactivity symptoms make a similar diagnosis in adulthood?

This work does not want to systematically review neurodevelopmental disorders, a very complicated enterprise, but to plant some seeds of reflection on these questions and associated consequences. New technical and methodological instruments (such as neuroimaging advanced techniques, but also refined statistical approaches such as machine learning) could be important tools in increasing our knowledge about psychopathological conditions. Still, they need to be driven by theoretical strength and flexible thinking. Otherwise, they can only amplify the existing problems.

## 2. Historical Background

Neurodevelopmental disorders currently include intellectual disabilities, communication disorders, autism spectrum disorder (ASD), ADHD, specific learning disorders (SLD), and movement disorders. The diagnosis of many of these disturbances underwent considerable redefinition during the 1900s, but only in the 21st century has there been a massive increase in research, a rapidly expanding literature, and significant media attention. For some neurodevelopmental disorders, there remains the question of whether they are recent onset conditions that appeared as sporadic cases in ancient times, such as anorexia, or disorders that have accompanied humans for several thousand years, such as schizophrenia [12].

In particular, SLD and ADHD are phenomena with a high resonance today, but they have appeared only very recently in human history.

The earliest descriptions of SLD date back to the late 1800s, when an ophthalmologist noticed the difficulty of some non-brain-damaged children in reading strings of words, a difficulty he christened “word blindness” [13]. Kirk first used the term “learning difficulty,” broadened to include other specific school difficulties encountered unexpectedly in children without cognitive retardation or behavioral disorders, in 1962 [14]. In any case, a true awareness of the extent of the problem and its effects on the child is new in recent decades. Over the past 25 years, interdisciplinary studies have traced the relationship between schooling, cognitive abilities, and associated comorbidities and have helped establish the basis for effective rehabilitative interventions. A Cochrane review on the efficacy of rehabilitative treatments of phonological has been conducted, concluding that they are efficient in improving some aspects of reading skills [15]. Many other studies carried out interventions or training based on other cognitive functions, such as attention, working memory, and visual-motor abilities [16,17]. Research into the neuroscience and genetics of SLD has benefited from modern technologies. As a result, the increased development in the scientific understanding of SLD has had significant implications for both assessment and intervention [18], with important implications for the organization of educational delivery. Bidirectionally, schools themselves may have contributed to the importance attached to SLD, thanks to the centrality that schooling has assumed in recent times. Several countries have indeed enacted interventions for inclusive schools, aligning with the intention of free and appropriate public school education for all children (such as the Office for Standards in Education, Children’s Services and Skills in UK, the Education for All Handicapped Children Act in the USA, and the statements on special education from the MIUR in Italy). In this general context, SLD has obtained special attention; the importance of providing best-practice guidelines for literacy teaching based on evidence has been supported by government-funded reports; for a review, see [19].

As with SLD, there is no clear description of ADHD in the past literature. A few isolated references to comparable disorders can be sporadically encountered in the European literature from the late 1700s to the early 1900s and in the United States from the early 1800s onward [20]. In the early decades of the 1900s, the term was usually associated with mental retardation of “child brain damage syndrome,” and researchers began to use it for precisely those children who today would be labeled ADHD [21]. That definition later evolved into the better-known term “minimal brain damage” and later “minimal brain dysfunction,” which were abandoned only in the 1980s, when the current nomenclature was first used in the DSM-III-R, with the three symptoms still recognized today as central criteria: inattention, hyperactivity, and impulsivity [22]. The development of the literature on ADHD was remarkable in the first decade of the 21st century; technology revolutionized ADHD research, associated, for the first time, with a rapidly expanding neuroimaging and genetics literature, as in the case of SLD. This literature has provided compelling new evidence for the biological correlates of ADHD and its complexity. ADHD seems highly heritable and multifactorial; multiple genes and non-inherited factors contribute to the disorder. Furthermore, prenatal and perinatal factors have been implicated as risks [23].

Filling in the historical gaps in the origin of the disorders is not just a historiographical exercise but could help us take a step forward in understanding the underlying causes; this, in turn, can help us understand the mechanisms that regulate the organization of the brain and may be the key to understanding its etiology and perhaps treatment.

## 3. Neurodevelopmental Disorders Today

Attention to neurodevelopmental disorders has gone hand in hand with their incidence, which appears to be steadily increasing. The debate regarding the causes of this increase is still open, but certainly, the increased attention to symptoms and better organization of the diagnostic process have played important roles. As an example, diagnoses of SLD have increased considerably internationally. In the USA, the National Center for Education Statistics reported that in 2015–2016, 6.7 million students (13% of all public school students) received special education. Among these, more were diagnosed with SLD than any other type of disability. The percentage of children with SLD shifted from 21.5% of all disabilities in 1976–1977 to 34.8% in 2014–2015, with a stable trend since the 80s [24]. For further study, reviews [25] and discussions [26,27] on the topic are available.

ADHD disorder is also estimated to have tripled in recent years. In the Italian context, the 2003 Consensus Conference in Cagliari marked the beginning of the National Registry and pharmacovigilance activities (officially started in 2007). The estimated prevalence, including clinical pictures of all levels of severity, is in the range of 0.4 percent to 3.6 percent [28], which is still lower than internationally.

Neurodevelopmental disorders, even if limited to specific learning domains such as ASDs, given their widespread occurrence, represent a high burden at the levels of health, society and economics, also because the actual management of the problem is not always linear [25,26,29]. In the absence of a clear organic correlate, diagnosis is based on collections of symptoms and often remains a subject of study and debate. The boundaries of many neurodevelopmental disorders appear blurred, multiple disorders and symptoms overlap, and manifestations are often atypical. Those who deal with these issues daily know that this is the rule rather than the exception.

Thus, as a new scenario has emerged, so have new problems. These problems probably originate from the nature of the healthcare system in industrialized countries, which was built around acute health problems [2]. This approach was successful when the goal was to reduce mortality. Currently, however, the most common condition is long-term coexistence with mild to moderate problems and chronic disorders, and the former treatment model is no longer optimal because it lacks a comprehensive view of the dynamics at work and the underlying mechanisms.

So it happens that despite great efforts to formalize the diagnostic process and understand many disorders’ etiology, pathophysiology and socio-environmental risk factors, actual diagnosis, intervention, and management of the problem can often be ineffective. For example, in a study in the United States, out of 50 pediatric services, only half of the professionals by their own admission had followed the guidelines in making the diagnosis of ADHD, and almost all (93%) reported prescribing medication immediately [30].

Relative age is a major determinant in ADHD diagnosis; younger children are up to 70% more likely than their classmates to be diagnosed with ADHD [31]. As we have discussed extensively in a previous paper, the possibility of interpreting behavioral immaturity as a disorder is a striking case of overdiagnosis and lack of awareness of developmental trajectories [26]. Maturation is characterized by physiological stages that are transient and unstable in children and can take on pathological significance if decontextualized. This happens if we embrace too tightly the rigid criteria of group averages and expected values by age, which have proven to be of little use for the individual, instead of assessing development from a developmental (lifespan) perspective. Such a perspective would also allow for greater continuity in managing many disorders that are currently viewed very differently at different life stages, with no seamlessness and handoffs at transitional stages (just think of patients who have been diagnosed with ASD as they enter adulthood).

In 2010, the U.K. government expressed concern about the expansion of diagnostic categories in DSM-5, under which normal variations in behavior were treated as diseases (Office for Standards in Education [32]). The same report denounced an over-identification of children with special educational needs. A practical consequence of such a way of operating is the stiffening of educational provisions, which is more oriented toward a dichotomous view of “normal-education” versus “special education,” dividing children into the two categories, and no longer embracing the challenge of an educational model that evolves in search of the right educational path for each child in its particularities.

Thus, positions on dealing with these disorders are not merely a theoretical debate that is consummated in the scientific arena but represent a real problem whose effects are have flow on effect on family, school, and health services.

### Comorbidities

Difficulties multiply for all those children with organic issues also present in comorbidities. In particular, in the presence of a neurological disease, there is a tendency to consider this the main cause and any other disorders mere “adjuncts.” In reality, the term comorbidity neither implies nor excludes a causal association and in fact, there is an over-representation of neurodevelopmental disorders.

A more 360-degree view is also taking hold in other areas of neurology, such as in the case of multiple sclerosis. The relationship between the neurological pathology and the other major symptoms, psychiatric disorders, cognitive disorders, and fatigue, is now clearly recognized and are all considered independent manifestations of the underlying demyelinating disease [33,34]. We have also had to report the coexistence of numerous disorders and a wide variety in the symptomatology presented when evaluating the development of children born prematurely [35] and with congenital heart disease [36,37]. Despite this, an integrated approach for these children is still not the norm in clinical practice.

The shift in perspective in favor of a deeper view of the disorders is therefore not limited to neurodevelopmental disorders but involves every area of medicine. However, there is still much internal resistance. For example, neurologists are not ready to handle psychiatric issues. Physicians, patients and family often overlook cognitive and psychiatric comorbidities, with disastrous consequences. Instead of being considered an obstacle in the diagnostic process of neurodevelopmental disorder, neurological pathology could be a resource for understanding brain functioning mechanisms (such as [38,39]), guiding us to a more complex and less modular view of psychic functioning.

## 4. Future Perspectives

As the focus has shifted from the acute medical problem to a more comprehensive view of the person or patient in a lifespan perspective, an obligatory step in the new medical and healthcare scenario, we have found ourselves moving onto shifting and unexplored terrain that has exposed many vulnerabilities in the previous system.

High interindividual variability is one of the most salient features of many neurodevelopmental disorders. In long-term outcomes of neurological pathology, it also occurs in groups of patients homogeneous in clinical characteristics, so much so that it can be considered a specific peculiarity of the pediatric patient. The developmental trajectories of higher cognitive functions depend on many unpredictable and closely interrelated variables. Individual characteristics, such as genetics, temperament, and specific vulnerability, interact with parental variables and disease, hospitalization, and medical procedures. The further one moves away from elementary cognitive functions, the more the variability increases. Individuals are remarkably diverse, showing variation across a range of behaviors and phenotypes; this is true in typical development but even more so in atypical development [40]. Moreover, cognition and personality take several years to develop; for this reason, the effects of an aversion event at a crucial developmental stage can often take years to manifest, a period when many factors can exert their influence.

Unfortunately, the lifespan perspective about developmental trajectories tends to reduce as one approaches adult age. In fact, there is again a great discontinuity and lack of communication between professionals dealing with childhood and adulthood. Longitudinal studies following the transition are always fewer, given the long-time requested to obtain results.

Long-term longitudinal studies are crucial in developmental cognitive neuroscience, for the inferential attributive process, and in understanding developmental trajectories. Today, the most advanced technologies would allow us to study the underlying neurofunctional mechanisms and see how they change with growth. This could help to understand the overlap between disorders better, as more symptoms may cross over to more disorders. This could also help us understand whether certain disorders really disappear in adulthood or if they take on a different guise as they mature and move into adulthood. This could have immediate consequences. For example, in the US, total stimulant usage for ADHD doubled in the last decade (also in children aged less than five years) [41]; in 2011, two-thirds of children diagnosed received pharmacotherapy [42]. How will this massive stimulant use evolve in the transition from childhood to adulthood? Is there a correlation with the remarkably increased benzodiazepine use recorded [43]?

The clinical management of people with a neurodevelopmental disorder and/or neurological disorder is certainly complex and involves several figures [4,44]. The great challenge for children at risk of developing cognitive and psychopathological difficulties is how to use the window of brain plasticity, which allows for the greatest learning possibilities. Treatment that is too focused on a single function may not be truly useful, and implemented interventions may work for some children but not be effective for all. Therefore, it is critically important to move beyond a specific symptom and begin to consider the impact of rehabilitative interventions as a whole. Instead, many issues are underestimated because of the focus of therapeutic attention on the most prominent symptom. This could be possible only if the system starts to work in a more integrated way [44]. For example, it is absurd that after an SLD diagnosis—often earlier—schools activate personalized educational programs consisting of dispensation/compensation methods. If associated with the absence of an effective rehabilitation program, this may mean a paradoxical loss of access to an adequate education. In fact, typical public schools and special education interventions often stabilize the degree of reading failure rather than remediate (i.e., normalize) reading skills [45]. To make the situation worse, treatments are highly resource-demanding, with very high costs for the families in terms of time and money. Therefore, not all children with SLD have access to rehabilitative programs or may not have prompt and continuous service [26].

The next challenge is to figure out how to use the mechanisms identified in rehabilitation programs to provide therapeutic care, whereby the benefits extend beyond specific tasks to aspects of daily life.

## Data Availability

Not applicable.

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
