# Peer review of "Neurodevelopmental Disorders: Past, Present, and Future"

_children, 2022, doi:10.3390/children10010031_

Round 1

Reviewer 1 Report

Thank you for submitting your paper to Children. It was a pleasure reading it. In my opinion it presents a new perspective to Neurodevelopmental Disorders that is interesting and appropriate. But the paper needs to be improved, so I suggest the following major revisions:

The central topic of the research article: Neurodevelopmental Disorders is of great interest although it has been approached from a very particular context and the results are not discussed with previous studies in this regard.

Once again, the topic is quite relevant, but it lacks more support and a better connection between the different sections of the document. In the introduction authors can explain their motivations and why the issue is relevant.

The introduction has few significant references. In some places it seems more like a manifesto or an opinion piece than an element of support for the arguments presented below. Once again, the topic is quite relevant, but it lacks more support and a better connection between the different sections of the document. In the introduction authors can explain their motivations and why the issue is relevant.

The literature review must be orderly: ideas are not connected. I suggest to group ideas and create a logic discourse.

In general, the feeling is that there was no clear objective for this paper, since it does not clearly present a methodology for studying, neither the results of a study with s strong scientific support. My suggestions would be for the authors to describe carefully all the procedures undertook and deeply assess the future Perspectives and the validity of the text itself.

Author Response

Thank you for submitting your paper to Children. It was a pleasure reading it. In my opinion it presents a new perspective to Neurodevelopmental Disorders that is interesting and appropriate. But the paper needs to be improved, so I suggest the following major revisions:

The central topic of the research article: Neurodevelopmental Disorders is of great interest although it has been approached from a very particular context and the results are not discussed with previous studies in this regard.

Once again, the topic is quite relevant, but it lacks more support and a better connection between the different sections of the document. In the introduction authors can explain their motivations and why the issue is relevant.

We really thank the reviewer for this comment. We enriched both the introduction section and the discussion, better explaining our motivations and the "file rouge" connecting all the sections of the work.

The introduction has few significant references. In some places it seems more like a manifesto or an opinion piece than an element of support for the arguments presented below. Once again, the topic is quite relevant, but it lacks more support and a better connection between the different sections of the document. In the introduction authors can explain their motivations and why the issue is relevant.

We agree; the background on which we based our argumentation has been enriched with several references.

The literature review must be orderly: ideas are not connected. I suggest to group ideas and create a logic discourse.

In general, the feeling is that there was no clear objective for this paper, since it does not clearly present a methodology for studying, neither the results of a study with s strong scientific support. My suggestions would be for the authors to describe carefully all the procedures undertook and deeply assess the future Perspectives and the validity of the text itself.

We thank the reviewer for the suggestions. We didn't want to do due a systematic review (also given the broad argument considered) but to suggest some insights about an argument in which we have been involved as health professionals for years. We better explained our reason and the scope of this work throughout the work.

Reviewer 2 Report

In the short literature review titled "Neurodevelopmental disorders: past, present, and future" the authors describe the recent shift regarding the societal perception and diagnosis of neurodevelopmental disorders, including ASD and ADHD. The authors then go on to illustrate how this progress has not been matched by improvement in and updating of the therapeutic approaches. The authors advocate in favour of a complete rethink of the medical system when it comes to paediatric cognitive disorders, including a wholistic approach.

This is a very interesting topic, particularly in the current post-pandemic context where young individuals have lost formative years in lockdown. Whilst the authors have done a solid effort in covering many aspects of the subject, it often remains shallow and superficial. For example, in the section referring to SLD, it would be useful to provide examples of assessment and interventions help ground the subject into a context. It is currently too vague to be informative.

Similarly, line 81/82: it would be extremely useful to describe some of the new evidences the authors are referring to.

Section 4 (Future perspectives) misses the point, it lacks direction. Specifically, I would recommend rewriting the sentences between line 174-176 as in the current form, it does not make sense. I would suggest making this final section more impactful, with clear recommendations as to which measures the authors believe would make a dramatic improvement in patients outcome/care/ or support their carers. For examples, parents of children with ASD, have to run from one therapist to another, usually after school when children are tired and not receptive. What is the authors opinion?

I found the section relating to diagnosis particularly interesting, specifically the over-identification and tendency to try and fit children in tiny little developmental boxes with very rigid age limits. And that, from the moment they are born. Do the authors believe this tendency is becoming more prominent?

The language is generally good, I only noted a few mistakes.

Line 25: change "his/her" for "their"

Line 30: change " this perspective" for "these changes"

Line 69: change "obvious" for clear or detailed

Line 101 : change "ranging" for "range"

Line 146  : change "declined" for "have flow on effect on" or "impact"

Author Response

In the short literature review titled "Neurodevelopmental disorders: past, present, and future" the authors describe the recent shift regarding the societal perception and diagnosis of neurodevelopmental disorders, including ASD and ADHD. The authors then go on to illustrate how this progress has not been matched by improvement in and updating of the therapeutic approaches. The authors advocate in favour of a complete rethink of the medical system when it comes to paediatric cognitive disorders, including a wholistic approach.

This is a very interesting topic, particularly in the current post-pandemic context where young individuals have lost formative years in lockdown. Whilst the authors have done a solid effort in covering many aspects of the subject, it often remains shallow and superficial. For example, in the section referring to SLD, it would be useful to provide examples of assessment and interventions help ground the subject into a context. It is currently too vague to be informative.

We thank the reviewer for the comments. We enriched the work to allow a more in-deep argumentation of the covered topics throughout the manuscript.

Similarly, line 81/82: it would be extremely useful to describe some of the new evidences the authors are referring to.

We have added some information and reference to this point.

Section 4 (Future perspectives) misses the point, it lacks direction. Specifically, I would recommend rewriting the sentences between line 174-176 as in the current form, it does not make sense.

The sentence has been reformulated as following:

The great challenge for children at risk of developing cognitive and psychopathological difficulties is how to use the window of brain plasticity, which allows the greatest learning possibilities.”

I would suggest making this final section more impactful, with clear recommendations as to which measures the authors believe would make a dramatic improvement in patients outcome/care/ or support their carers. For examples, parents of children with ASD, have to run from one therapist to another, usually after school when children are tired and not receptive. What is the authors opinion?

We thank the reviewer for the comment. We enriched the discussion following the suggestions of the reviewer.

I found the section relating to diagnosis particularly interesting, specifically the over-identification and tendency to try and fit children in tiny little developmental boxes with very rigid age limits. And that, from the moment they are born. Do the authors believe this tendency is becoming more prominent?

Yes, we think that this is the risk of having all this effort in early diagnosis and in the tendency to consider “pathological” every “difference”. We tried to make our opinion more clear throughout the manuscript.

The language is generally good, I only noted a few mistakes.

Line 25: change "his/her" for "their"

Done

Line 30: change " this perspective" for "these changes"

Done

Line 69: change "obvious" for clear or detailed

Done

Line 101 : change "ranging" for "range"

Done

Line 146  : change "declined" for "have flow on effect on" or "impact"

Done

Round 2

Reviewer 1 Report

The text can be accepted in the present form